



# The MAESTRO turbulence dataset derived from the SAFIRE ATR42 aircraft

Louis Jaffeux[1], Marie Lothon[1], Fleur Couvreux[4], Dominique Bouniol[4], Grégoire Cayez[3], Lilian Joly[5], Jérémie Burgalat[5], Cyrille De Saint Leger[3], Hubert Bellec[3], Olivier Henry[3], Dyaa Chbib[3], Tetanya Jiang[3], and Sandrine Bony[2]

[1]Laboratoire d'Aérologie, University of Toulouse, CNRS, UPS, Toulouse
[2] LMD/IPSL, CNRS, Sorbonne University, Paris, France
[3] SAFIRE, CNRS, Météo-France, Toulouse, France
[4] CNRM, Météo-France, Toulouse, France
[5] GSMA, Reims-Champagne Ardenne University, Reims, France

**Correspondence:** Louis Jaffeux (louis_jaffeux@gmx.fr) and Marie Lothon (marie.lothon@cnrs.fr)

**Abstract.** The MAESTRO airborne field campaign took place between August 10 and September 10 2024 over the North Atlantic tropical ocean near the Cabo Verde Islands. Its goal was to investigate the processes that control the mesoscale organization of clouds with a payload of probes and sensors, as well as vertically and horizontally-pointing radars and lidars. A particular attention was paid to the role of coherent structures in the boundary layer and mesoscale cloud organization. This focus motivated the acquisition of high-resolution measurements of temperature and water vapor to capture turbulence dynamics in the subcloud layer. To achieve this, six hygrometers and four temperature sensors were deployed, including a new fast-rate hygrometer called FAST-WAVE. This article describes the turbulence dataset, prepared on the basis of these measurements. It consists in 25 Hz segmented time series of calibrated water vapor mixing ratio, temperature, and three-dimensional wind, their corresponding fluctuations, as well as turbulent moments, and integral length scales. In total, 40 hours of stabilized legs data were gathered in a wide range of mesoscale and local cloud conditions, with nearly 13 hours consisting of high-quality boundary-layer samples. This paper describes the methodological choices made for all the computations, calibrations, and corrections that were applied to the original measurements. The collection of NetCDF files composing this dataset is publicly available on the AERIS website.

## 1 Introduction

The role of clouds in global climate dynamics remains associated with many mysteries and open questions. The simulation of clouds in numerical models also remains challenging, especially the representation of the diversity and behavior of low-level clouds over the oceans (Konsta et al., 2022). The tropical ocean is populated by many cloud types, ranging from small shallow cumulus to deep convective clouds. Low-level clouds are ubiquitous in this environment and their albedo makes them key players in the energy budget of the Earth. These clouds interact with their environment by exchanging latent heat through evaporation and condensation, and by absorbing solar radiation. Using observations, several configurations have been described in the boundary layer with respect to convection, turbulence, and cloudiness. For example, shallow convection has





been linked to organized structures in the subcloud layer, such as roll vortices creating cloud streets (LeMone and Pennell (1976)). In another case, the turbulence in the subcloud layer was found to be locally suppressed through cloud precipitation or evaporation while a cold pool formed, enhancing turbulent fluxes in the boundary layer at its front (de Szoeke et al., 2017).

In another configuration the boundary layer and the clouds above were decoupled (Nowak et al., 2021). Understanding how such scenarios interplay, influence local circulation, contribute to broader cloud organization patterns, and, finally, affect the radiative impact of clouds, relies on the acquisition of representative observations.

Tropical marine clouds inherently exist at the intersection of mesoscale organized patterns and chaotic variability, where large-scale circulations and external drivers, such as the warm tropical ocean, supply the energy that generates turbulence.

Starting with the GATE experiment, scientists have formulated hypotheses about the organization of tropical clouds for decades (Warner et al., 1979), and recently systematic classifications of cloud organization using satellite images have been proposed over the tropical ocean (Stevens et al., 2020; Janssens et al., 2021). Links between the variety of cloud patterns and near-surface wind and stability of the low troposphere were pointed out (Bony et al., 2020) and confirmed by ground based instrumentation over Barbados (Schulz et al., 2021). Data from the EUREC4A field campaign (Bony et al., 2017, 2022) were used to understand

these different patterns (Bony et al. in prep). Using LES simulation, Dauhut et al. (2023) reproduced and analyzed one case of meso-scale cloud organization observed during EUREC4A and highlighted the importance of cold pools and mesoscale circulation in its formation. Other LES studies used this dataset to evaluate whether current models are able to reproduce cloud organizations (Schulz and Stevens, 2023; Jansson et al., 2023). As is often the sign of success in scientific research, EUREC4A answers questions by raising new ones, that have already inspired the community to return to the field and collect new data.

From August to October 2024, a network of international field campaigns, named ORCESTRA (Organized Convection and EarthCARE Studies over the Tropical Atlantic), took place over the tropical Atlantic to investigate the organization of convection, namely MAESTRO (Mesoscale organisation of tropical convection), PERCUSSION (Persistent EarthCare underflight studies of the ITCZ and organized convection), BOW-TIE (Beobachtung von Ozean und Wolken – Das Trans ITCZ Experiment), PICCOLO (Process investigation of clouds and convective organization over the tropical ocean), CLARINET (CLoud

and Aerosol Remote sensing for EarThcare), CELLO (Cloud and EarthCARE caL/vaL Observations), SCORE (Sub-Cloud Observations of Rain Evaporation), and STRINQS (Soundings and TuRbulent eddy measurements in the ITCZ with a Network of QuadcopterS) projects. A common objective of these campaigns, is to collect relevant observations in the atmosphere in the vicinity of the Inter-Tropical Convergence Zone (ITCZ). These eight research projects therefore gathered around this focal point under the ORCESTRA umbrella to plan simultaneous field campaign deployments, first in the eastern, then in the west-

ern North Tropical Atlantic Ocean. This collaboration fostered numerous instrumental synergies and joint missions, including coordination with spaceborne operations, as well as encouraging long-term research cooperation that will build on the data collected during the campaigns.

Within ORCESTRA, the MAESTRO field campaign used the instrumented ATR-42 research aircraft from SAFIRE to investigate the physical processes governing the mesoscale organization of shallow and deep convection. This campaign follows the

earlier mentioned 2020 EUREC4A field campaign and aims to pursue the scientific questions that arose from it. The interplay between coherent structures in the sub-cloud layer and the mesoscale organization of clouds is one of the primary objectives

of MAESTRO. A combined analysis of airborne measurements of cloud properties and spatial distributions, and in-situ turbulent scales and fluxes in the subcloud layer, was envisioned to address this issue. State-of-the-art turbulence measurements are therefore a main requirement for the completion of the objectives of MAESTRO. Retrievals of proper turbulent fluctuation

time series and moment computations require expertise in instrumentation, data filtering, and processing.

The goal of this article is to present the collected data, to describe the associated workflow, and to share them with the scientific community. This dataset includes high-frequency calibrated time series and turbulent fluctuation time series for temperature, 3D-wind, and water vapor mixing ratio, as well as corresponding turbulent moments and associated errors. Section 2 describes the context of the campaign, the instrumentation used for in-situ turbulence measurements, and the sampling strate-

gies. Section 3 presents the data processing, including improvements made to the original high-rate data, in particular for the wind corrections and the calibrations of high-rate temperature and humidity sensors, as well as the computation methodology for fluctuations, turbulent moments, and integral length scales. Section 4 presents an overview of the final products and discusses the strengths and limitations of the dataset, as well as possible future improvements. Finally, section 5 concludes the article with recommendations for the users of the dataset and its future within the MAESTRO project.

## 2  The MAESTRO field campaign : acquisition strategy

This section provides a brief overview of the MAESTRO campaign, outlining its objectives, the instrumentation used for turbulence measurements, and the adopted sampling strategies.

### 2.1  Description of the campaign

The MAESTRO project aimed to 1) test hypotheses for mechanisms explaining the mesoscale organization of shallow and

deep convection, 2) investigate whether and how this organization influences the Earth's radiation budget, 3) provide an extensive dataset to evaluate high resolution models in their prediction of mesoscale cloud-related organizations. To achieve these objectives, the field campaign targeted mesoscale conditions ranging from shallow to deep convection. Various cloud types and their organizations were systematically observed, with consistent sampling of the marine boundary layer, cloud base level, and mid-troposphere. The campaign documented several phenomena and cloud organizations ranging from cloud streets, con-

vergence lines, cold pools, isolated short-lived convective cells up to large mesoscale convective systems (MCS). Operating from Sal Island, Cabo Verde, the MAESTRO field campaign consisted in 24 airborne missions performed between August 10 and September 10 2024. The ATR-42 research aircraft from SAFIRE was equipped with a payload of in-situ and remote sensing instruments, including horizontally and vertically pointing radars and lidars. The high-rate in-situ measurements of temperature, 3-dimensional wind, and water vapor, collected during the 24 MAESTRO research flights (later referred to as

RF), amount to more than 40 flight hours of stabilized legs corresponding roughly to 15 000 km of air sampled. The projected flight trajectories are shown in Figure 1. The south-eastern part of the Cabo Verde area was preferably sampled as the air masses originated from the east, thereby promoting air unaltered by land, and the southern region was closest to the northern





edge of the ITCZ which is a region of key interest for the ORCESTRA community. In the vicinity of Sal airport, from four to six soundings were performed daily to characterize the whole depth of the troposphere.

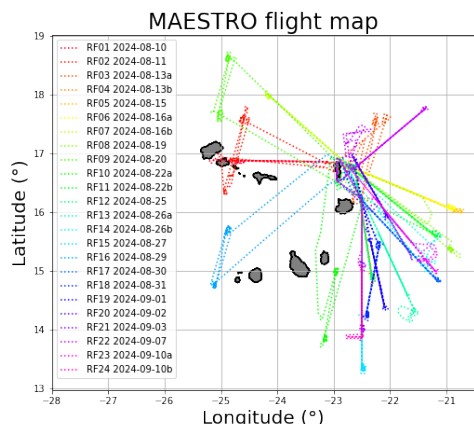

**Figure 1.** Trajectories of the 24 MAESTRO Research Flights (RF) of the ATR-42. Each flight is named RFXX (MAESTRO nomenclature) or 2024-MM-DD-a or -b (ORCESTRA nomenclature).

## 2.2 Instrumentation for turbulence measurements

The ATR-42 from SAFIRE is an instrumented airplane equipped with 4 high-rate temperature probes, and 6 hygrometers. Horizontal and vertical wind components are derived from measurements taken by a five-hole radome positioned at the nose of the aircraft, Pitot air speed sensors, and an inertial navigation unit (INS), following the equations of Brown et al. (1983). Although some of this instrumentation acquires data with rates exceeding 100 Hz, the high-rate measurements were resampled to a common 25 Hz frequency consistent with EUREC4A (Brilouet et al., 2021), where the same aircraft and similar instrumentation were used. Given an aircraft ground speed of 100 m.s$^{-1}$, 25 Hz is equivalent to a 4 m spacing between each measurement. The hygrometers used included :

- WVSS-2 based on laser absorption spectroscopy with acquisition frequency of 0.5 Hz.

- the newly developed FAST-WAVE (Fast Aircraft Spectrometer for Tracking Water Vapor Emissions), based on laser absorption spectroscopy with acquisition frequency of 100 Hz.

- the Li7500 (Licor), based on near infrared absorption, with a 50 ms response time, and 50 Hz acquisition frequency.

- 1011 C, a chilled mirror measuring the dew point temperature with a response time of 1 °C/sec and acquisition frequency of 1 Hz.

- HUMAERO and UCAP, two calibrated capacitive sensors sensitive to relative humidity and associated with their own temperature sensors, with acquisition frequencies of 1 and 40 Hz, respectively.





The temperature measurements were performed using two platinum wire thermometers (deiced and non-deiced E102AL Rosemount), two fine wire probes with wire diameters of 5 and 10 $\mu$m, measuring impact temperatures faster than 100 Hz (Ballantyne and Moss, 1977). More information on the thermodynamic ATR intrumentation is available in part 3.1 of Bony et al. (2022).

## 2.3 Flight sampling strategy

The MAESTRO campaign was dedicated to the gathering of an extensive dataset, representative of various convective clouds and their surrounding environments ranging from shallow, low-level cumulus clouds to deeper convective systems. To achieve this, a series of stacked horizontal legs at predefined flight levels were planned and executed. A typical MAESTRO flight pattern consisted of a sequence of 200 km-long transects, designed as follows:

- A single H-type (High) transect at mid-tropospheric altitude ( 6000 meters above sea level), used for downward-looking remote sensing and in situ measurements of ice clouds and air motion.

- Two B-type (Base) back-and-forth transects at cloud base, enabling both in situ and remote sensing observations of the cloud base region, with particular emphasis on cloud–subcloud thermal interactions.

- One final L-type (Low) transect below the clouds dedicated to in situ measurements within the subcloud layer and upward-looking remote sensing.

The sequence order was sometimes modified, placing the high-altitude transect last. This change improved fuel efficiency by taking advantage of the lighter weight of the aircraft during ascent, enabling higher altitudes to be reached and additional legs to be performed within the spared flight time. Finally, S-type (Surface) legs, were sometimes conducted depending on the conditions. They were flown exclusively during day time at about 60 meters above the sea surface to document turbulence in the surface layer. However, potential alteration of other instruments due to exposure to sea salt was suspected, in particular for the horizontally pointing Lidar. For this reason, these legs were flown only on a few occasions during the campaign. An example of a typical MAESTRO flight plan is given in Figure 2.

Along this primary flight plan, the campaign also featured flights coordinated with other projects that were part of OR-CESTRA. On seven flights (RF4, RF7, RF11, RF12, RF15, RF16, and RF21), the german DLR HALO aircraft, operating as part of the PERCUSSION campaign, deployed drop-sondes in a circular pattern around the operational area of the ATR-42. This enabled estimates of mesoscale vertical velocity within the region of interest, similarly to what was done during EUREC4A (Bony and Stevens, 2019; George et al., 2021). During RF18 and RF16, coordination with the King Air aircraft from the CELLO project was performed with both airplanes flying in close formation, either side by side or one above the other. These coordinated operations not only completed the measurements performed on a given day, offering exceptional case study opportunities, but also made it possible to compare simultaneous remote sensing and in-situ techniques used by the different platforms. Flights RF02, RF03, RF09, RF10, and RF18 were synchronized and aligned along the satellite track of the newly launched EarthCARE satellite (Wehr et al., 2023) for the calibration and validation of its onboard remote sensing instruments.

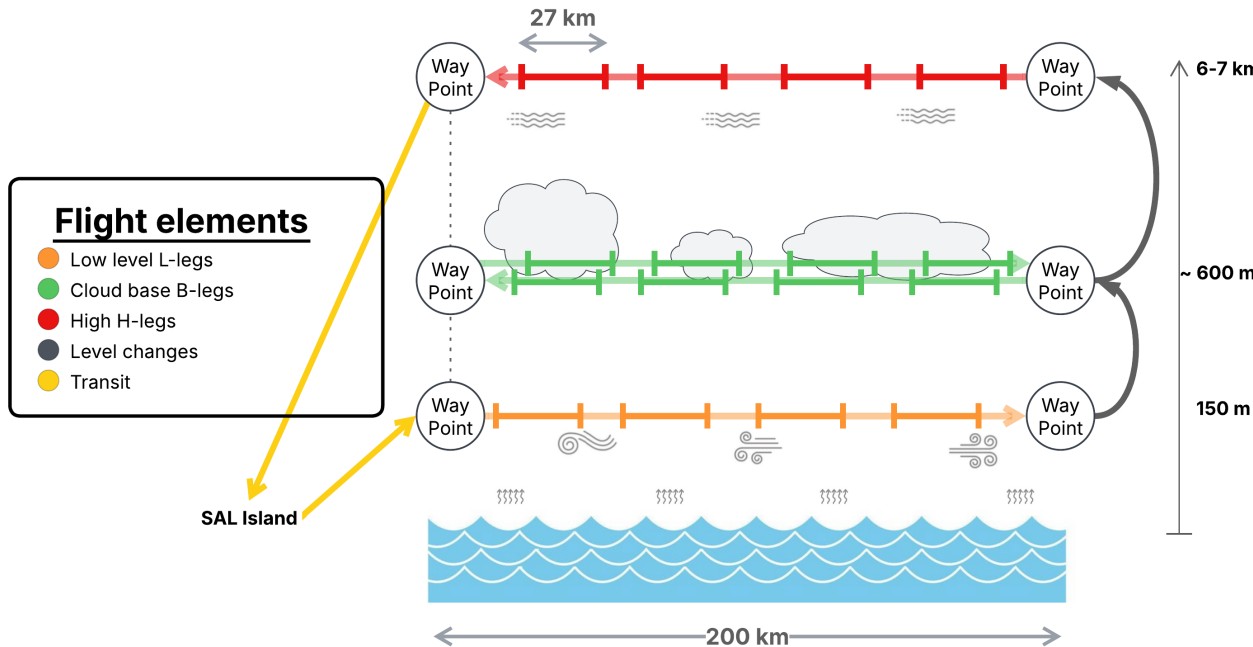

**Figure 2.** Schematic profile of a typical MAESTRO flight plan. Single arrows symbolize flight elements, such as stabilized legs, transit, or ascents. Line segments bounded by orthogonal end bars symbolize flight segments defined with the segmentation algorithm.

During flights RF04, RF05, RF07, RF12, RF15, RF19, and RF20, RCM and Sentinel1 satellites acquired Synthetic Aperture Radar images on the aircraft track. SAR is sensitive to sea surface roughness, especially centimeter-scale waves. To take advantage of this, the flight plans were adjusted so that low-level legs (S- or L-type) coincided with the satellite overpasses, providing unique opportunities to investigate sea surface–atmosphere interactions. Finally, in the event of exceptional convective activity in the vicinity of Sal Island, flight plans were adapted to avoid dangerous areas and prioritize regions of highest scientific interest.

The flight coordinated with EarthCARE varied slightly from the typical flight plan described in Figure 2, but kept its main elements as much as possible. A few other flights (e.g RF19 and RF20) used different flight plans adapted to specific cloud organizations. However, the nomenclature used to name each transect is similar, allowing the data from all MAESTRO flights to be merged in a single, coherent dataset. To characterize the additional leg types that were not part of the typical strategy, new categories were defined :

– M-type (mid-level) legs, corresponding to intermediate altitudes, between 1 and 4 km.

– T-type (top) legs, corresponding to low-cloud tops, between 1 and 1.5 km.





These legs enabled upward and downward pointing of remote sensing instrumentation to characterize cloud geometry, as well as the collection of in-situ microphysical and thermodynamical measurements in specific regions. This general leg nomenclature is common to all MAESTRO datasets.

For the turbulent dataset, the legs undergo an automatic segmentation algorithm, creating 4.5-minute segments ($\approx 27$ km)
by minimizing the heterogeneity of all 5 turbulent fluctuations simultaneously. This quantification allows the identification of sufficiently homogeneous segments for reliable turbulent moment estimates. To do this an heterogeneity score $\mathcal{H}$ has been defined as follows:

$$
\mathcal{H} = \sum_{v \in [U,V,W,T,q]} \sqrt{\int_0^{\Delta T} \left( 100 \cdot \frac{\int_0^t \left(F^{(v)}(x)\right)^2 dx}{\int_0^{\Delta T} \left(F^{(v)}(x)\right)^2 dx} - 100t \right)^2 dt}
\tag{1}
$$

where $F^v$ refers to the fluctuation time series for the variable $v$ and $\Delta T$ is the segment duration. $\mathcal{H}$ represents the average percentage deviation from an ideal uniform accumulation of variance over the time series summed over all five variables. In particular, the cumulative variance ($\int_0^t \left(F^{(v)}(x)\right)^2 dx$) is scaled as a percentage of the total variance over the segment ($\int_0^{\Delta t} \left(F^{(v)}(x)\right)^2 dx$) and compared to a linear rate of accumulation, calculating the root-mean square error between the two functions. Finally, the resulting averaged percentage error is summed for all five variables to obtain $\mathcal{H}$. The algorithm to automatically optimize segment placements is described in the appendix Figure A1. This algorithm ensures that the maximum number of segments is obtained, and sequentially minimizes $\mathcal{H}$ (i.e. maximizes the homogeneity of corresponding time series).

Initially, the segment length was defined as 5 minutes ($\approx 30$ km) following the recommendations of Lenschow et al. (1994) for reliable turbulent moment estimations. Since some legs were shorter than this initial threshold, this length was adjusted to maximize data usage while maintaining a consistent duration for each segment, easing their comparison. A quantitative summary of the obtained segments, categorized by flight strategies and leg types, is presented in Table 1.

| Strategy | Flight list (RF) | Number of Flights | S | L | B | H | T | M | Total |
|---|---|---|---|---|---|---|---|---|---|
| **Typical MAESTRO** | 4, 5, 6, 7, 8, 11, 12, 13, 15, 17, 23 | 11 | 3 | 78 | 131 | 68 | 0 | 0 | 280 |
| **Case study** | 14, 19, 20, 21, 22, 24 | 6 | 0 | 67 | 26 | 22 | 6 | 24 | 145 |
| **Coordination** | 1, 2, 3, 9, 10, 16, 18 | 7 | 6 | 44 | 60 | 28 | 0 | 0 | 138 |
| **Total** | | 24 | 9 | 189 | 217 | 118 | 6 | 24 | 563 |

**Table 1.** Summary of flights and segments of each type, sorted by strategy.

Figure 3 presents the histograms of $\mathcal{H}$ calculated for segments of each category throughout the entire MAESTRO campaign. As expected, low-level segments are generally more homogeneous than those collected at higher altitudes. Using an upper threshold of 30% to define homogeneity (green shaded area in Figure 3), most S-type and L-type segments qualify as homogeneous and are suitable for boundary-layer turbulent moment computation. This results in a total of 170 homoge-

neous boundary-layer segments (totalling all S-,L-, and B-type segments with $\mathcal{H}$<30%) included in the MAESTRO turbulence

175    dataset.

Segments with $\mathcal{H}$ values between 30% and 60% (blue shaded area in Figure 3) can be considered of intermediate levels of heterogeneity. These likely contain local-scale structures, such as thermals and clouds, which explains their predominance among B-type segments. Such structures are important to characterize for the broader MAESTRO campaign. These segments may still be used for analysis, but the moment estimates are less reliable.

180    Segments with $\mathcal{H}$ values exceeding 60% (red shaded area in Figure 3) should not be used for turbulence statistics, as such calculations rely on the assumption of homogeneity. These high heterogeneity scores likely indicate the influence of more laminar flow, non turbulent, heterogeneous or intermittent turbulence large-scale dynamics, consistent with the observation that they predominantly occur in H-type segments collected at high altitudes.

As discussed in Section 3.3.5, the associated random errors increase significantly above the 30% heterogeneity threshold,

185    resulting in high uncertainty in turbulence moments and integral length scales.

In the case of L-type and S-type legs, high heterogeneity scores typically result from encountering localized atmospheric events such as cold pools, rainfall, or particularly stable layers.

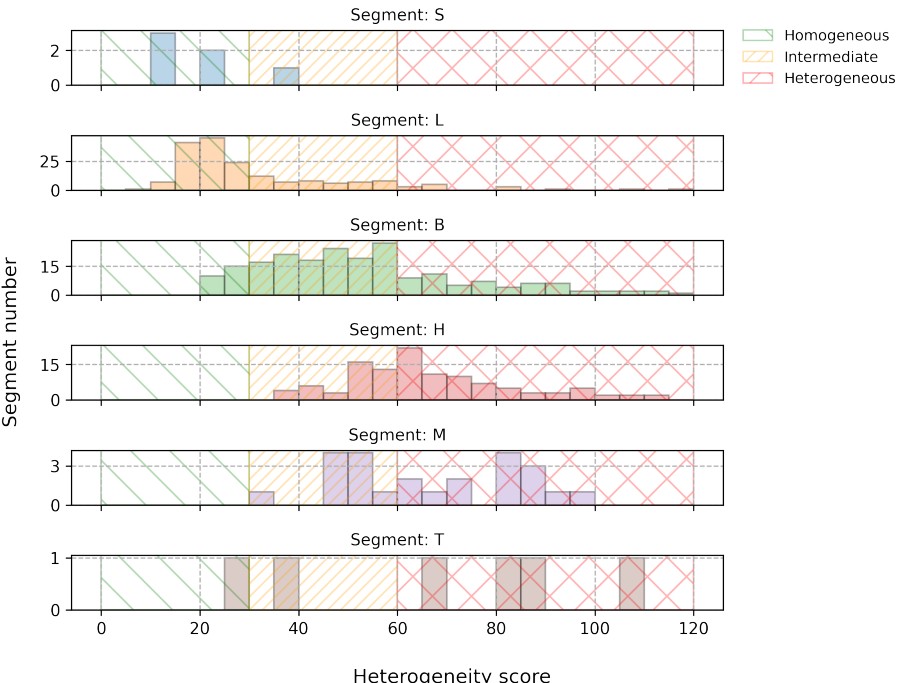

**Figure 3.** Histograms of $\mathcal{H}$ (Heterogeneity score) for each segment type. Shaded regions highlight qualitative thresholds: scores below 30% (green) indicate homogeneous segments, while those above 60% (red) relate to heterogeneous segments. Scores between 30% and 60% (orange) represent an intermediate heterogeneity range, where segments may be suitable for analysis but warrant cautious interpretation.



## 3 Data processing

This section presents the processing workflow and discusses how instrumental redundancy is leveraged to ensure the quality of the MAESTRO turbulent dataset. It describes the calibration and correction methodologies applied to the five variables used for the extraction of turbulent features, along with the rationale behind these adjustments. Finally, an evaluation based on amplitude and spectral behavior analysis is presented to determine the most suitable sensor for temperature and humidity measurements and to validate the overall data quality.

### 3.1 Processing workflow

#### 3.1.1 Computation of fluctuations

Turbulent fluctuations $x'(t)$ are defined, for any given variable $x(t)$, as :

$$x(t) = \overline{x(t)} + x'(t) \tag{2}$$

where $\overline{x(t)}$ is the mean component of $x(t)$. There are multiple ways to compute this mean component and thereby extract the corresponding fluctuations, including segment average, detrending and high-pass filtering. For the MAESTRO Turbulence dataset, this is achieved by applying a high-pass filter with a cutoff frequency of 0.018 Hz, equivalent to filtering mesoscale features larger than approximately 5 km, following step by step the methodology presented in Brilouet et al. (2021). Because the use of frequency cutting filters involves arbitrary choices, in particular the numerical method used to approximate the Fourier transform, alternative fluctuations using a straightforward linear detrending method are also computed and provided in the dataset.

Once segmentation and pre-processing have been performed, the turbulent fluctuations at 25 Hz are computed for each segment and for the 5 thermodynamical variables $U_L$, $V_T$, $W$, $\theta$, and $q$, where $U_L$ is the horizontal stream-wise wind component, $V_T$ the horizontal transverse wind component, computed by projecting the geographical wind components onto the mean wind at the segment scale, $W$ is the vertical air speed, $\theta$ the potential temperature, and $q$ the water vapor mixing ratio. Figure 4 shows an example of such time series for a sub-cloud layer leg of RF06. In addition to this set of variables, fluctuations of the horizontal wind components are also provided in the geographical frame of reference (zonal and meridional wind components) and in the frame of reference of the plane.

#### 3.1.2 Computation of turbulent moments and integral length scales

The 25 Hz fluctuations are used to compute turbulence moments. Second order moments provide information about the energy contained in the fluctuations. For their computation, the eddy correlation method is used, so that for any two variables $x$ and $y$,



**Figure 4.** Fluctuation time series from segment L2-1 of RF06 for $U_L$, $V_T$, $W$, $\theta$, and $q$, respectively.

and for segments of duration $\Delta T$ :

$$\overline{x'y'} = \frac{1}{\Delta T} \int_{0}^{\Delta T} x'(t)y'(t)\,dt \tag{3}$$

   Figure 5 shows a computed autocorrelation function, its integral, and the spectrum derived from a segment time series. Integral length scales are obtained as the integral of the autocorrelation function for the high-pass filtered fluctuation or cross correlation up to its first crossing with the x-axis (Lenschow et al., 1994) (Figures 5a and b). The integral length scale represents

a measure of the size of the largest eddies in a turbulent flow and indicates the average length over which turbulent structures
are sustained before they start to transfer energy to smaller scale eddies in the Kolmogorov energy cascade (Figure 5c).

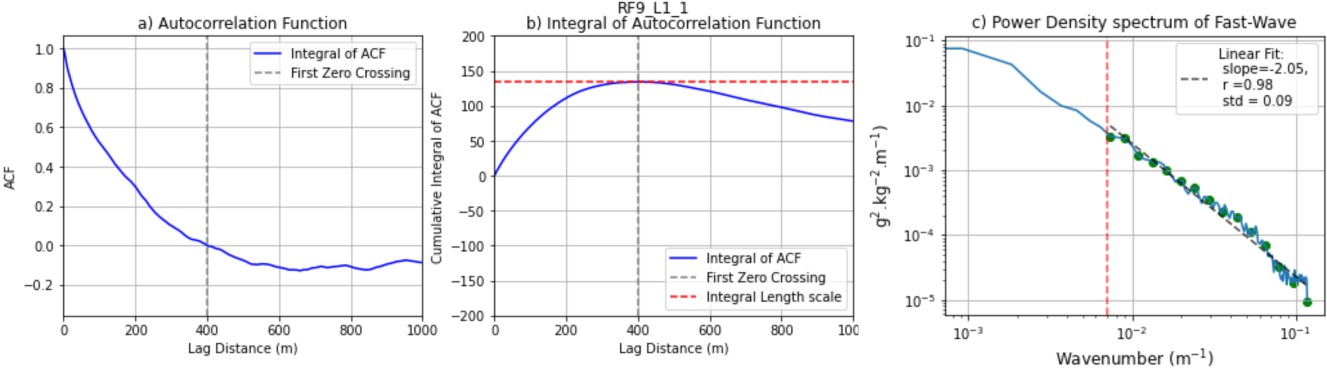

**Figure 5.** Example of autocorrelation function, its integral, and obtained spectrum for the water vapor mixing ratio measured with FAST-WAVE for segment L1-1 of RF09.

The TKE dissipation rate $\varepsilon$ is also computed. It measures the rate at which turbulent kinetic energy is converted into heat
through viscous effects at small scales. Because this parameter governs the energy budget, it is an important result of the
analysis of the turbulent flow. It can be estimated with different numerical methods. In this case, it is computed assuming
Kolmogorov's scaling from power spectra such as the one shown in Figure 5c, using the formula of Lambert and Durand
(1998) for the vertical wind $w$ in the inertial subrange which is defined beyond the integral length scale and up to angular wave
numbers corresponding to 25 Hz ($\approx 1.6$ rad.m$^{-1}$, at 100 m.s$^{-1}$). Consequently, $\varepsilon$ is defined as:

$$\varepsilon = \sigma_f^3 [2\alpha(k_{IL}^{-2/3} - k_{max}^{-2/3})]^{-3/2} \qquad (4)$$

where $k_{IL}$ and $k_{max}$ are the wave numbers corresponding to the vertical wind integral length scale and the upper bound of the
wavelength spectrum, respectively, $\sigma_f$ is the variance of the integrated vertical wind spectrum between $k_{IL}$ and $k_{max}$, and $\alpha$
is the Kolmogorov coefficient, assumed equal to 0.52. This formulation shows that $\varepsilon$ is directly linked to the velocity gradients
and is higher in regions of strong velocity variations and shear. In atmospheric modeling, TKE dissipation rate determines the
energy dissipation at the subgrid scale and is therefore required for closing the Navier-Stokes equation. The values obtained
for the MAESTRO dataset are showed in Figure 17 and described in Section 4.

**3.2  Pre-Processing**

As most field campaigns, MAESTRO was an opportunity to develop and test new instrumentation, including a new high-rate
hygrometer : FAST-WAVE. This instrument was developed after the Atmospheric Measurements by Ultra-Light SpEctrometer
(AMULSE) technology by the GSMA (Groupe de Spectrométrie Moléculaire et Atmosphérique) (Joly et al., 2016), based on
wavelength modulation spectroscopy. It was housed in a certified antenna, which was mounted on the airplane fuselage. It is

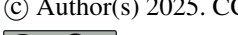



the first time that such an hygrometer is used onboard a research aircraft. To validate the instrumental payload and specifically test this sensor prior to the field missions, 4 test flights were performed in the South West of France during the spring of 2024. These flights helped ensure the quality of instrument integration on the aircraft, data treatment procedures, and acquisition protocols. The subsequent analysis raised important concerns about the wind retrieval with unusual values for averaged angles of attack and sideslip, resulting in non negligible averaged vertical wind. For long enough period, the averaged vertical wind

is expected to vanish and be very close to $0\pm0.1$ m.s$^{-1}$. This is usually used as a criterion in the calibration of airborne wind measurements (Mallaun et al., 2015; Khelif et al., 1999). This hypothesis was generally valid during the straight legs of the MAESTRO campaign. However, significant correlation (25%) was noted outside of these legs. The current subsection details the train of thought that led to corrections of the airplane wind, and the calibration methodology for temperature and water vapor mixing ratio.

### 3.2.1    Wind corrections

The air velocity components are obtained by subtracting the ground velocity of the plane, measured by the INS, to the air velocity with respect to the aircraft, deduced from the differential pressures measured by the five-hole nose radome and a Pitot tube. Aircraft flight inevitably disturbs the air flow, and therefore all sensing probes mounted on any aircraft are not directly measuring the meteorological undisturbed airflow. To mitigate this effect, gust probes are often mounted on the nose of the

aircraft and calibrated, which is the case for the 5-hole radome of the ATR-42. In addition, there are multiple potential sources of errors that include uncertainty about the positions and alignments of the INS and radome nose, and instrument performance. In the case of turbulence studies, high accuracy in these measurement is of primary importance.

    Calibration maneuvers were performed during RF22, on 7th of September 2024, as described in Lenschow (1986) to confirm and resolve the apparent wind measurement issue. In particular, a reverse heading maneuver was performed by flying at constant

height and true heading starting with the wind in the back, then taking a 90-degree heading change to the right, after that taking a 180-degree heading change to the right, and finally taking a 90-degree turn to the left. This maneuver was performed in 27 minutes at an altitude of 2.2 km, outside of the boundary layer. The initially computed wind during the reverse heading maneuver is shown in Figure 6 (panels a and b). The basic hypothesis behind this maneuver is that the horizontal and vertical winds should be independent of the attitude parameters of the plane. In particular with different heading mean values, during

the first and last straight segments and during the second and third one, the retrieved 3D-wind should be nearly identical.

    The initially obtained horizontal wind (Figure 6a) strongly varies in direction and strength when crossing the same air volume after a 90 and 180 degrees heading change. Moreover the vertical wind shows a strong positive change as the plane turns to the right, and a strong negative change as the plane turns to the left (see Figure 6b). This indicates that the vertical wind is heavily correlated with the roll of the aircraft.

The adopted approach to compute biases in both radome angles using the data from stabilized legs consists in a parameter optimization (PO) that uses the wind computation formulas and optimizes two biases for the angles of attack and sideslip to minimize the vertical wind averaged for the whole campaign. This methodology is similar to the ones adopted by the German Aerospace Center DLR (Giez et al., 2021; Mallaun et al., 2015). However, rather than determining a set of biases for each



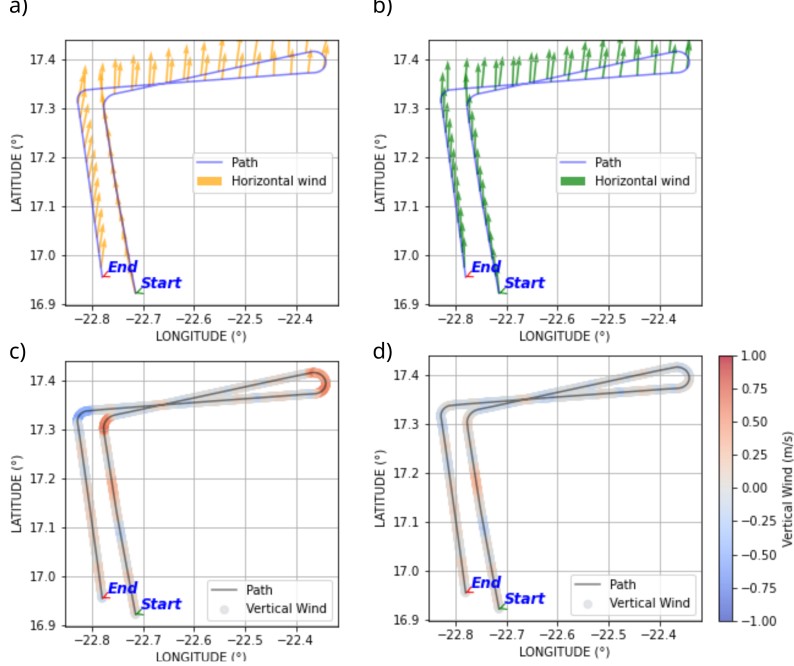

**Figure 6.** Computed wind vectors during the reverse heading maneuver with and without wind corrections. Initially computed horizontal (a) and vertical (c) winds along the plane's trajectory. Corrected horizontal (b) and vertical (d) winds along the plane's trajectory. This maneuvers took place during RF22 on September 7, 2024.

flight, the method was used once, at the scale of the whole campaign assuming a non drifting sensor defect common to the
entire dataset. The other hypothesis used in these studies, namely the minimization of the correlation between the vertical
wind and the roll angle of the plane, was not required. Instead, simultaneous optimization of both angle biases reduced this
correlation to negligible levels, from 25 to 1.4 %. The calculations produced biases of -0.015 and -0.545 degrees for the attack
and sideslip angles, respectively.

  The wind has then been computed again, taking into account (subtracting) the obtained biases for both measured incidence
angles. The horizontal wind corrected along the plane trajectory for the reverse heading maneuver (Figure 6c) shows a direction
and strength close to identical as the airplane travels through the same air volumes. The vertical wind along the trajectory
(Figure 6d) does not show any significant increase during the right turns or decrease during the left turn, demonstrating that
roll and vertical wind are now uncorrelated. The main flaws that were initially identified in both the horizontal and vertical
wind fields have thus been corrected by taking into account two fixed biases for the angle of attack and sideslip that were
estimated using the full campaign dataset. The adopted correction will affect both the mean wind fields and the extracted
turbulent moments and retrieved spectra. The latter is evaluated in the following subsection 3.3.


### 3.2.2   Calibrations of temperature and humidity instruments

Fast humidity measurements have long been a challenge for airborne platforms, because fast sensors are usually calibrated in controlled environments at the ground and experience much harsher conditions during aircraft operations. The methodology
applied here and taken from Brilouet et al. (2021), relies on the synergy between fast and slow reference sensors, allowing frequent calibrations of the fast sensor with a reference below a common low frequency, accounting for possible instrumental drifts. The calibration methodology is applied for each segment and for each fast sensor relative to each reference sensor mentioned in Section 2.2. Figure 7 illustrates this methodology with an example for FAST-WAVE during RF06. It consists in four steps:

1. Low pass filtering of the fast and slow sensor with a 1/6 Hz cutoff frequency (Figure 7a).

2. Optimal lag search to maximize correlation between the two filtered signals (Figure 7b).

3. Linear regression between the fast and low-pass filtered signals (Figure 7c).

4. Application of the retrieved slope and offset to the original fast signal (Figure 7d).

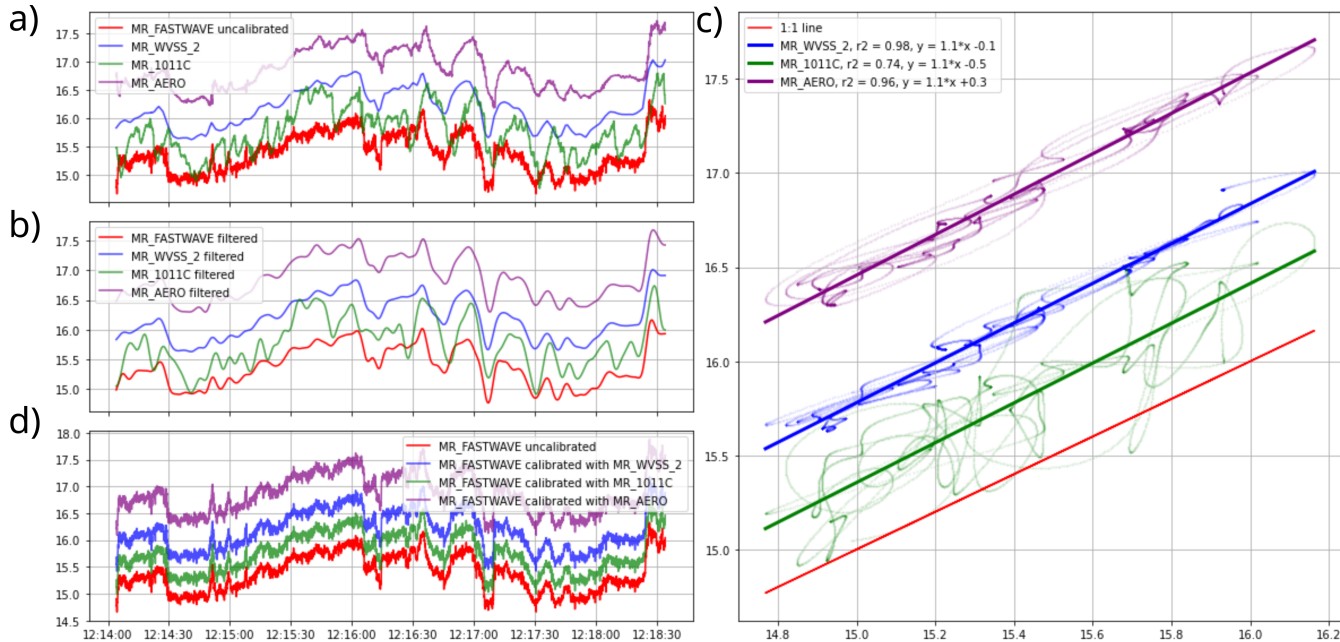

**Figure 7.** Example application of the calibration methodology taken from RF06 : a) uncalibrated time series for FAST-WAVE and the slow reference sensors (in g/kg) b) corresponding low-pass filtered time series, c) linear regression plots for each reference (yaxis) with respect to FAST-WAVE, d) calibrated FAST-WAVE time series for each reference.

For a given segment and fast instrument, $R^2$ values are compared to identify the optimal reference sensor for acquiring the
highest quality signals. Following the initial methodology, correlation coefficients exceeding 0.90 indicate successful calibra-



tions. This approach significantly helped monitoring potential instrument degradation throughout the campaign. Coupled with a spectral analysis of high-rate sensors, the best fast and reference pair of sensors are picked for high frequency temperature and humidity measurements to be used in the whole dataset, based on the correlation coefficients and the characteristics of the power spectra. This analysis is presented in the following subsection 3.3.

### 3.3 Post-Processing and Quality Analysis

#### 3.3.1 Amplitude and Spectral Domain Quality Analysis : Choice of best sensors

On the one hand, the corrections of winds and the calibrations of temperature and water vapor mixing ratio modulate the amplitude of the fluctuations and thereby significantly influence the extracted turbulent moment computations. For example, a calibration slope of 1.1 will increase the variance by 21%, in comparison with a 1.0 slope. Therefore, ensuring the validity of the employed procedures is crucial. On the other hand, the spectral behavior of the parameters should be compatible with Kolmogorov's turbulence theory. The current subsection explores both aspects with respect to each variable measured during the campaign, in order to qualify the processed data : first for water vapor mixing ratio, then for temperature, and finally for the three wind components.

#### 3.3.2 Water Vapor Mixing Ratio

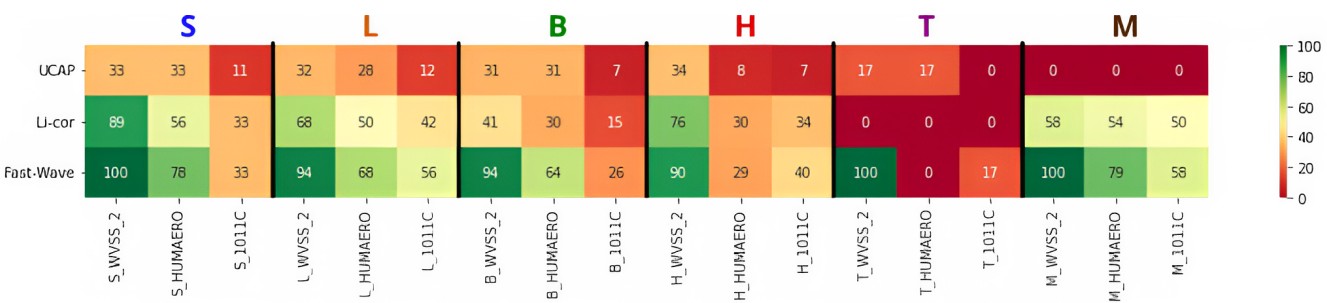

**Figure 8.** Results of the fast moisture sensor calibration. Percentage of segments with fast-slow sensor low frequency linear regression coefficient $R^2$ values above 0.9 for each segment category (heights) and each fast and reference humidity sensor.

The outcome of the rapid humidity calibrations is presented in Figure 8, categorized by segment type and reference sensor. FAST-WAVE demonstrates the best agreement across all segment types and reference sensors, with only minor exceptions in a few T and H segments. Among the reference sensors, WVSS2 consistently shows the best agreement across all segment types and fast sensors, except for the T-segments, where UCAP aligns better with HUMAERO than WVSS2. Overall, the results demonstrate that FAST-WAVE and WVSS2 are the most reliable combination for humidity calibration, achieving effective performance across 94% of the segments. This highlights their robust compatibility and suitability for diverse segment types, with only very few exceptions. In addition to the calibrations, a quality analysis of the spectral behavior of each humidity sensor was performed on the dataset to determine which sensor best matched the expected theoretical turbulence spectrum. For





each segment, the spectrum of the signal is computed using Fast Fourier Transform (FFT). Then, the wavelengths greater than the integral length scale are resampled to be equally spaced on a logarithmic scale and, together with the corresponding energy values, they are used to perform a linear regression in log-log space over the such-defined inertial sub-range (as illustrated in Figure 5c). Several important metrics can be obtained from this operation : 1) the slope which should be as close as possible to -5/3 for a turbulent sample, and 2) the correlation coefficient which indicates the match with the turbulent cascade deduced from the Kolmogorov Similarity hypothesis. The inertial sub-range definition used here implies that significant noise levels would result in lower slope values and low correlation coefficient. For the three hygrometers, the two spectrum parameters are displayed in Figure 9 as a 2D representation for each of the three statistically significant segment types. Comparatively, the slopes obtained with FAST-WAVE are the closest to the expected -5/3 value for all segment types. The correlation coefficient is also more often above the 0.9 threshold. Only for some type-L segments, UCAP provides better coefficients than FAST-WAVE, demonstrating its potential as a fast-rate instrument, however with less reliability. Overall, FAST-WAVE demonstrates the best spectral behavior among all high-rate hygrometers. As a result of both the amplitude and spectral comparisons, we concluded that, for the MAESTRO dataset, FAST-WAVE, calibrated using the WVSS-2, is the best sensor choice to use for humidity measurements.

### 3.3.3 Temperature

The results of fast temperature calibrations are displayed in Figure 10 for each segment category and reference sensor, in the same way as for fast moisture sensors in Figure 8. The non deiced temperature sensor is more in line with the fast instruments for every segment type, except for the M and T categories. The two fine wires show similar performance, with an edge to the 5 $\mu$m one. Overall, the results demonstrate that 5 $\mu$m fine wire and non deiced temperature sensor are the most reliable combination for temperature calibration, achieving effective performance across 79% of the segments. This highlights their robust compatibility and suitability for diverse segment types, with only minor exceptions.

Similarly to what was done with the humidity sensors, a quality analysis of the spectral behavior of each temperature sensor was performed on the dataset to determine which sensor matched best the theoretical expected turbulence spectrum. For the analysis, the non deiced Rosemount was included as it is also a fast sensor, with acquisition frequency greater than the required 25 Hz. The estimated correlation coefficient and slopes are shown in Figure 11.

Following the two criteria, the non deiced Rosemount probe demonstrates the best behavior for any segment type despite the two fine wires being specifically designed for fast rate measurements with effectively faster response times and frequencies. This non-deiced probe also presents the advantage not to require further calibration, which adds uncertainty in the turbulent moments estimates. We chose to consider this reference fast sensor as the best sensor choice for temperature fluctuation measurement.

### 3.3.4 3D Wind

To statistically assess the effect of wind corrections, the variance computed on each leg of the campaign was compared before and after correction. This comparison is summarized in the form of histograms in Figure 12. The variance of the horizontal

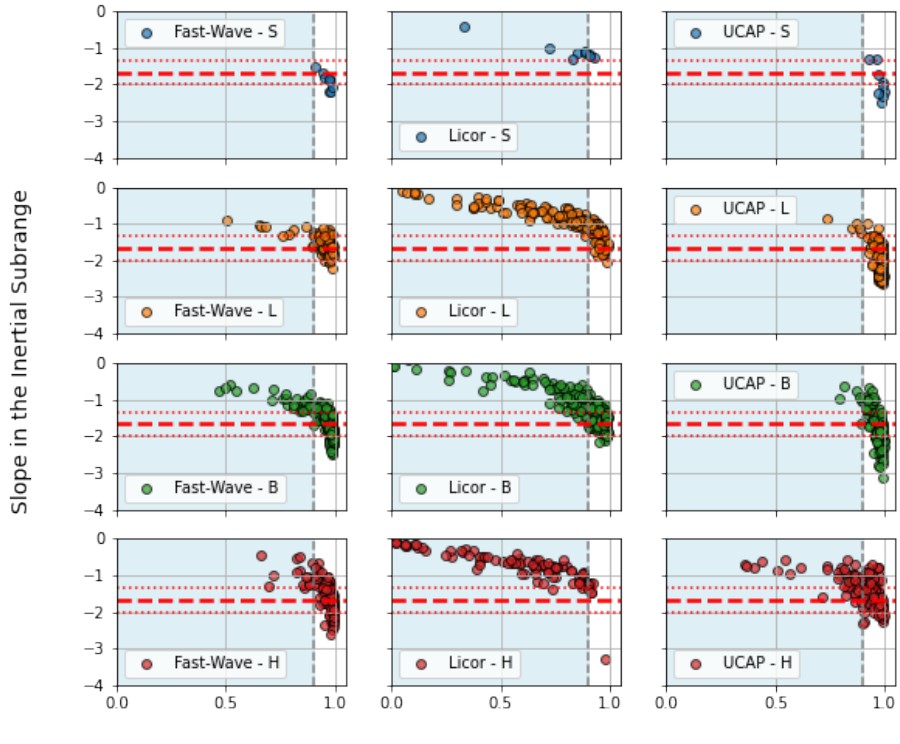

**Figure 9.** Spectral analysis of humidity sensors : results of linear regressions by instrument and segment type. The light blue area covers correlation coefficients below 0.9, the red dotted lines indicate slope values of -2 and -4/3, the red dashed line represents the -5/3 slope expected from classical turbulence theory.

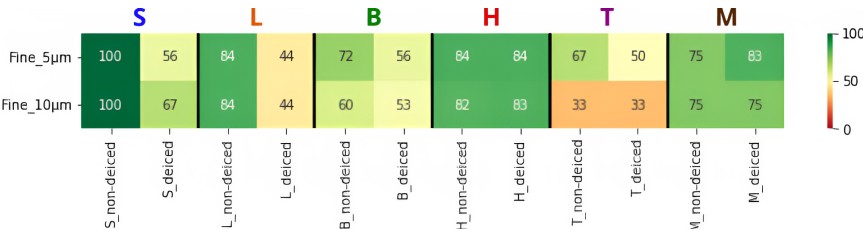

**Figure 10.** Results of the temperature calibration. Percentage of segments with $R^2$ values above 0.9 for each category and each fast and reference temperature sensor.

components of the retrieved wind vector see significant change from the radome corrections. Variance variations above 20%


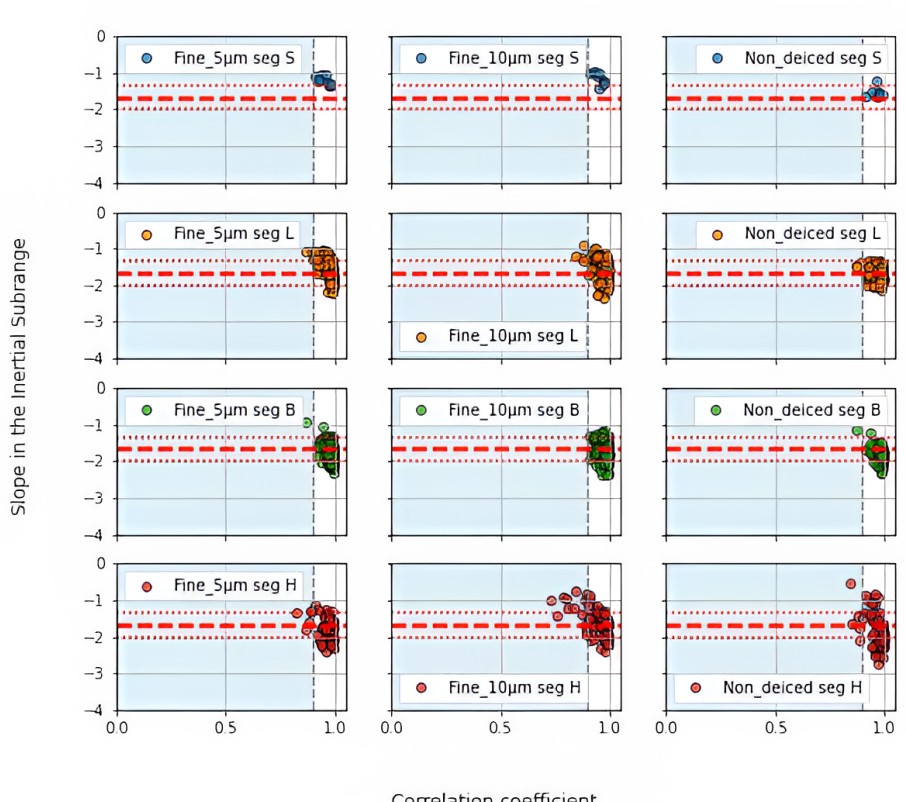

**Figure 11.** Spectral analysis of humidity sensors : results of linear regressions by instrument and segment type. The light blue area covers correlation coefficients below 0.9, the red dotted lines indicate slope values of -2 and -4/3, the red dashed line represents the -5/3 slope expected by the theory of turbulence.

are reached mostly in B and H segments. The distributions of the ratios of variance between streamwise and transverse components are symmetric for every segment types. The TKE and vertical wind variance, however, are not affected by the radome corrections. Because TKE is half the sum of the 3 variances, it follows that the main effect of the corrections is a redistribution

of the variance between the two horizontal components depending on the wind direction relative to the airplane heading.

To enable a direct comparison with previously presented results for temperature and water vapor mixing ratio, the spectral characteristics of the wind field are also examined. Figure 13 presents the power spectral density (PSD) distributions for each wind component. The analysis is conducted in a manner consistent with the methodology applied to the scalar fields. Across all segments classified as surface-layer (S-type) and lower-boundary-layer (L-type), the derived spectral slopes and corresponding

correlation coefficients exhibit strong agreement with theoretical predictions, particularly with the well-established -5/3 power-law scaling in the inertial subrange. This consistency suggests that turbulence in these regions exhibits the expected quasi-

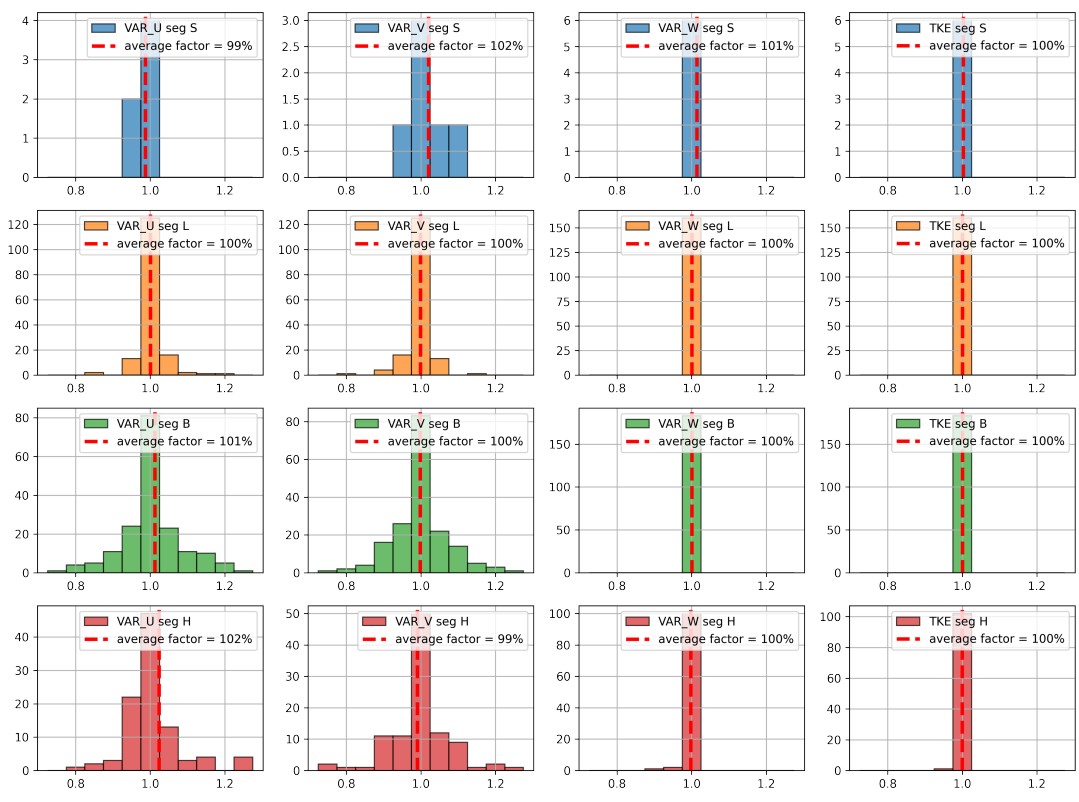

**Figure 12.** Histograms of the ratio before and after correction for the variance of the three wind components and the turbulent kinetic energy for S, L, B, and H segments.

isotropic cascade behavior. At higher altitudes, progressing into the upper boundary layer and free troposphere, the spectral slope and correlation coefficient values demonstrate increased variability for all three wind components. This broadening in parameter space reflects the transition to more laminar, or large-scale dynamical regimes, where turbulence becomes weaker, or increasingly intermittent.




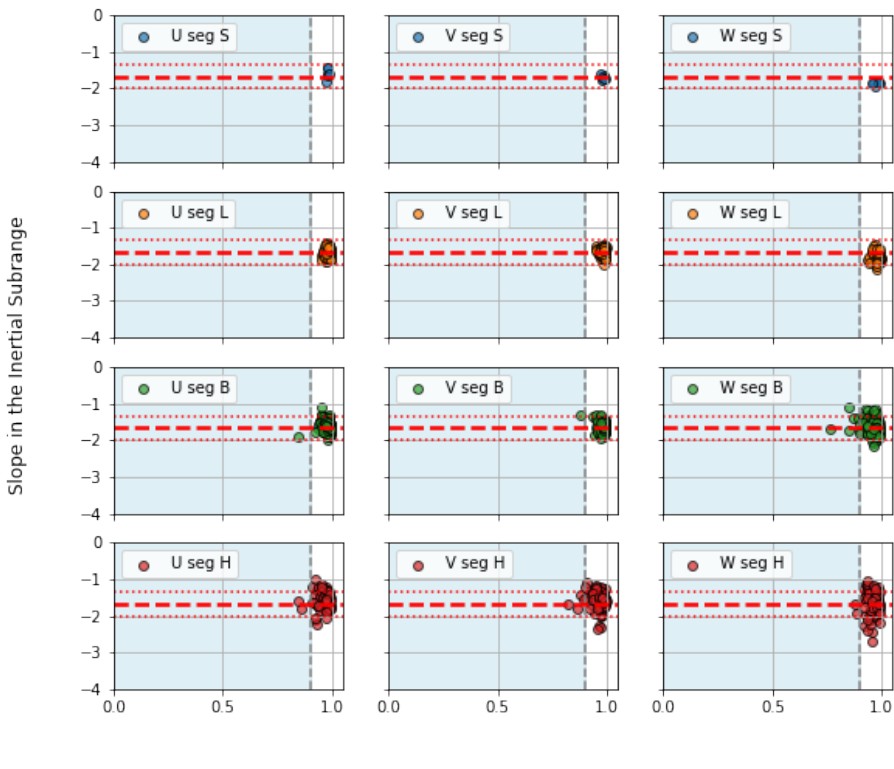

**Figure 13.** Spectral analysis of wind components by segment type. The light blue area covers correlation coefficients below 0.9, the red dotted lines indicate slope values of -2 and -4/3, the red dashed line represents the -5/3 slope expected by the theory of turbulence.

### 3.3.5 Errors associated with turbulent moments

### 3.3.6 Random error

Beyond instrumental limitations, the reliability of turbulent moment estimations depends on the sampling and filtering conditions. Following Lenschow et al. (1994), each of these effects can be accounted for by computing random and systematic errors, respectively. The random error ($\epsilon_r$) is inherent to the sampling conditions, in particular the finite length of the sample used to compute the turbulent moments relatively to the corresponding integral length scales. For any two variables $x$ and $y$, the random error associated with their covariance can be estimated as:

$$\epsilon_r = \sqrt{\frac{2L_{xy}}{L}}\sqrt{1 + \frac{1}{r_{xy}^2}} \tag{5}$$



where $L_{xy}$ is the integral length scale of $x'y'$, $L$ the length of the used segment, and $r_{xy}$ is the correlation coefficient between

$x$ and $y$. For the variance of a single variable $x$, the random error is defined as:

$$\epsilon_r = 2 \times \sqrt{\frac{L_x}{L}} \qquad (6)$$

In both cases, obtaining reliable estimates (low values of $\epsilon_r$) requires the sample length to be significantly larger that the integral length scale ($L >> L_{xy}$ or $L >> L_x$, respectively). Figure 14 shows the histograms of random errors associated with the variances of all 5 variables for the whole campaign and for each type of segments. All 5 variances exhibit similar trend with

respect to the 6 segment types. Variance estimates are relevant mostly for boundary-layer categories. For the three boundary-layer categories (S, L, and B), most of the segments show variance random error below 20% and positively skewed distributions. This positive skewness suggests that while the majority of random error values remain low—indicating sufficiently large sample sizes for reliable estimates—there is a minority of segments that suffered sampling limitations. Interestingly, the B category, which was expected to be highly heterogeneous (as illustrated in Figure 3) due to its diverse conditions—such as intermittent

cloud encounters, in-cloud segments, and out-of-cloud segments—displays a single-mode distribution of random errors. This is surprising, given that such variability should result in significantly different integral length scales and therefore different level of random errors giving a distribution with several modes. For the H category, which are the only statistically significant population of segments outside the boundary layer, the random error distributions are negatively skewed and centered around values ranging from 20 to 30%. Most segments exhibit high random errors, with a few segments still achieving lower error

values. Such a pattern is consistent with sampling conditions in the free atmosphere, where larger integral length scales are usually obtained, leading to relatively high random errors. The few samples with smaller integral length scales indicate that in a few cases, turbulence might still be at play.

### 3.3.7 Systematic error

As mentioned in Subsection 3.1.1, the definition of the mean field and consequently, the method used for its computation and

calibration, directly influences the magnitude of the fluctuations. The systematic error stems from the loss of information due to the choices made for computing fluctuations, in this case high-pass filtering. To estimate this loss, the covariance of the high-pass-filtered series ($F_{fil}$) is compared to the covariance of the detrended series ($F_{det}$):

$$\epsilon_s = \frac{F_{det} - F_{fil}}{F_{det}} \qquad (7)$$

These errors are computed for every second order moment, including variances and covariances, and are provided in the

dataset in order to give an estimate of the removed variance due to scales larger than 5 km. Figure 15 shows the histograms of systematic errors associated with the variances of all 5 variables for the whole campaign and for each type of segments. For the three boundary-layer segment types (S, L, and B), the histograms are very different in the case of vertical wind variance relatively to the case of horizontal components, with a much narrower distribution positioned entirely below 25%. This is explained by the very little effect large scales have on the vertical wind in the turbulent mixed layer. For the other 4 variables,



**Figure 14.** Histograms of random errors associated with the variances of all 5 variables for the whole campaign and for each type of segments. Colours indicate heterogeneity class of corresponding segments.

even in the boundary layer, much more variance is usually removed with the 5 km scale filter, revealing the relatively higher influence of the meso-scale on these thermodynamic parameters.

## 4 Dataset summary

As an illustration of the turbulence measured during the MAESTRO campaign, Figure 16 presents histograms of the variances, computed from the high-pass filtered series for each segment category. The instruments used to compute these moments are

the one selected for the dataset, after corrections and calibrations (see subsections 3.2 and 3.3). For the three wind components, the variance is usually higher within the boundary layer, in particular in L-segments. These high variance values indicate more turbulent wind behavior near the ground, which is characteristic of the atmospheric boundary layer. More precisely, it can be related to thermal convection or shear-driven turbulence, for example. For temperature and water vapor mixing ratio, cloud

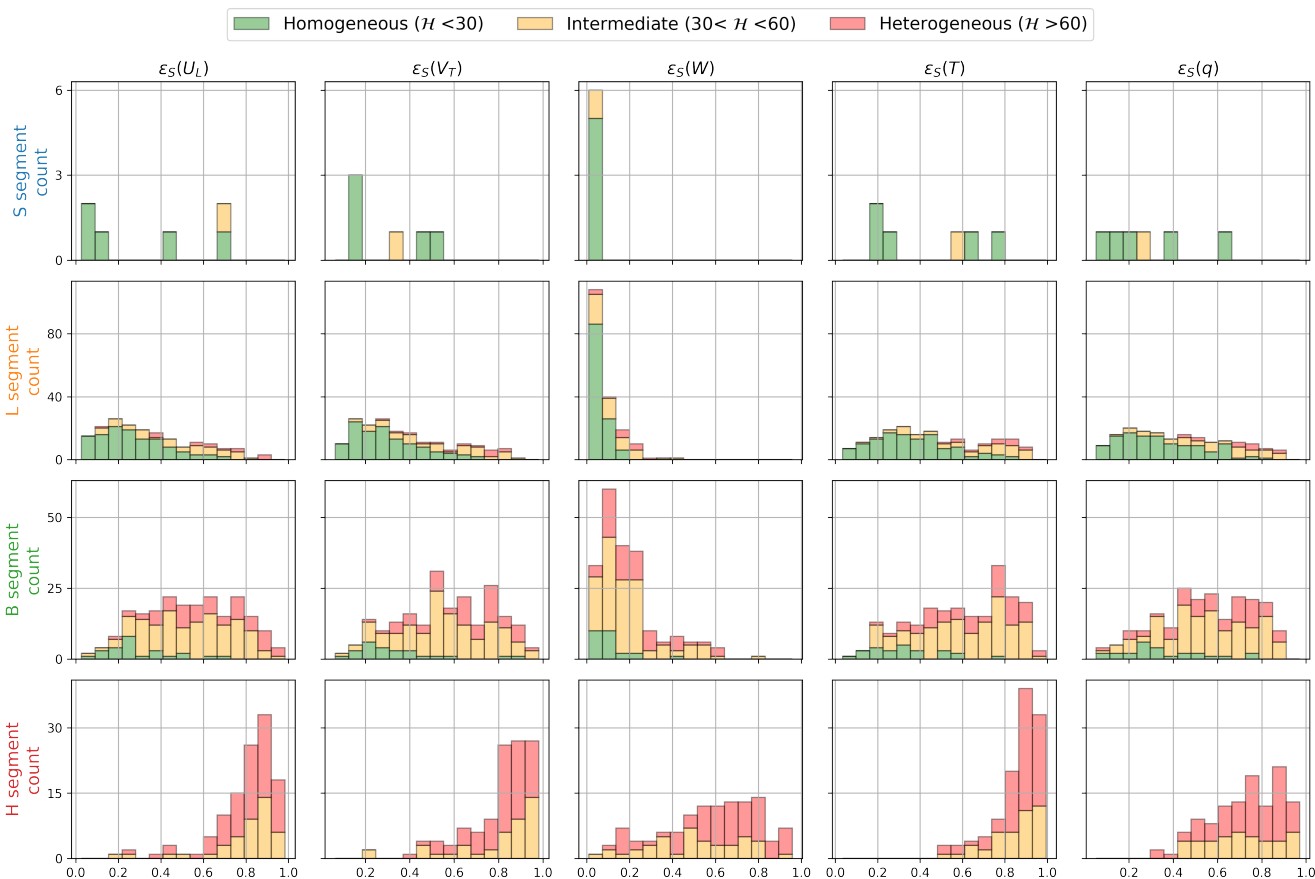

**Figure 15.** Histograms of systematic errors associated with the variances of all 5 variables for the whole campaign and for each type of segments.

bases (B-type segments) are associated with the highest variance values. The usually sharp gradients in temperature and hu-

midity at cloud base contribute to enhanced fluctuations, driven by latent heat release and vertical mixing. Some of these higher variance values can therefore be attributed to the crossing of this inversion in the case of relatively high heterogeneity segments. However, they may also be indicative of processes related to the dynamics of clouds such as condensation, evaporation, and entrainment.

Figure 17 shows the TKE dissipation rate measured during MAESTRO. The general trend is an increase towards the sea

surface (segments L and S) whose roughness contributes to turbulence. The obtained values for boundary-layer segments (L and S types) are in-line with the one that can be found in the literature (for example in Lambert and Durand (1998)) ranging between $10^{-4}$ and $10^{-3}$ m$^2$.s$^{-3}$. For higher altitudes, the values decrease substantially as flow regimes become more laminar and less energetic.

The dataset is divided by flight, and for each flight three kinds of NetCDF files are generated:



**Table 2.** List of variables calculated over each segment, their standard notation, and NetCDF variable names.

| Category | Notation(s) | NetCDF variable name(s) |
|---|---|---|
| **General characteristics** | | |
| Segment name | | name |
| Segment start/end | $t_i, t_f, t$ | time_start, time_end |
| Segment start/end/mean position | $lat_i, lon_i, lat_f, lon_f, lat, lon$ | lat_start, lon_start, lat_end, lon_end, lat, lon |
| Mean altitude | $z$ | alt |
| Mean heading | $THDG$ | MEAN_THDG |
| Mean ground speed | $GS$ | MEAN_GS |
| Mean true airspeed | $TAS$ | MEAN_TAS |
| Mean static pressure | $P_S$ | MEAN_PS |
| Mean geographic wind | $U_{geo}, V_{geo}$ | UWE_mean, VNS_mean |
| Heterogeneity score | $\mathcal{H}$ | H_score |
| Mixing ratio correlation coefficient | $R^2_{MR}$ | R2_MR |
| Horizontal wind speed and direction | $V_H, DD$ | WS_MEAN, MEAN_WD |
| **Turbulent quantities** | | |
| Second-order moments | $\overline{U_L'^2}, \overline{V_T'^2}, \overline{W'^2}, \overline{\theta'^2}, \overline{q'^2}$ | VAR_U, VAR_V, VAR_W, VAR_T, VAR_MR |
| Third-order moments | $\overline{U_L'^3}, \overline{V_T'^3}, \overline{W'^3},$ $\overline{\theta'^3}, \overline{q'^3}$ | M3_U, M3_V, M3_W, M3_T, M3_MR |
| Skewness | $S(U_L'), S(V_T'), S(W'),$ $S(\theta'), S(q')$ | SKEW_U, SKEW_V, SKEW_W, SKEW_T, SKEW_MR |
| Turbulent fluxes | $\overline{U_L'V_T'}, \overline{U_L'W'}, \overline{V_T'W'},$ $\overline{U_L'\theta'}, \overline{U_L'q'}$ $\overline{V_T'\theta'}, \overline{V_T'q'}, \overline{W'\theta'},$ $\overline{W'q'}, \overline{\theta'q'}$ | COVAR_UV, COVAR_UW, COVAR_VW, COVAR_UT, COVAR_UMR COVAR_VT, COVAR_VMR, COVAR_WT, COVAR_WMR, COVAR_TMR |
| Integral Length scales | $L_{U_L}, L_{V_T}, L_W, L_\theta, L_q$ | L_U, L_V, L_W, L_T, L_MR |
| Cross-correlation scales | $L_{U_L V_T}, L_{U_L W}, L_{V_T W}, L_{U_L \theta}, L_{U_L q},$ $L_{V_T \theta}, L_{V_T q}, L_{Wq}, L_{W q}, L_{\theta q},$ | L_UV, L_UW, L_VW, L_UT, L_UMR L_VT, L_VMR, L_WT, L_WMR, L_TMR |
| Turbulent kinetic energy | $E$ | TKE |
| TKE dissipation rate | $\epsilon$ | TKE_DR |
| Inertial range spectrum slopes | $s_{U_L}, s_{V_T}, s_W, s_\theta, s_q$ | Slope_U, Slope_V, Slope_W, Slope_T, Slope_MR |



**Figure 16.** Histograms of computed variance for each segment category and the 5 high-pass filtered variables.

1. Segment-scale moments (one file per flight). The turbulent moments and associated errors, along with segment-averaged values for each thermodynamic variable (referenced in Table 2), are provided for each segment. Additionally, the starting, and ending coordinates, mean true heading, true airspeed, and ground speed for a given flight are included.

2. Segment-scale time series (one file per segment). For each segment, one file is generated that includes the 25 Hz geographical coordinates, the corrected horizontal wind components in the geographic, stream-wise, and plane's frame of reference, the corrected vertical wind component, the calibrated water vapor mixing ratio, and potential temperature. In addition, fluctuations of the horizontal stream-wise wind components, vertical wind component, water vapor mixing ratio, and potential temperature are provided for high-pass filtered data, as well as detrended series.

3. Leg-scale time series and moments (one file per leg). Files similar to the two previous categories are also generated at the scale of entire legs, without constraints on homogeneity. The purpose of this approach is to provide a suitable dataset for



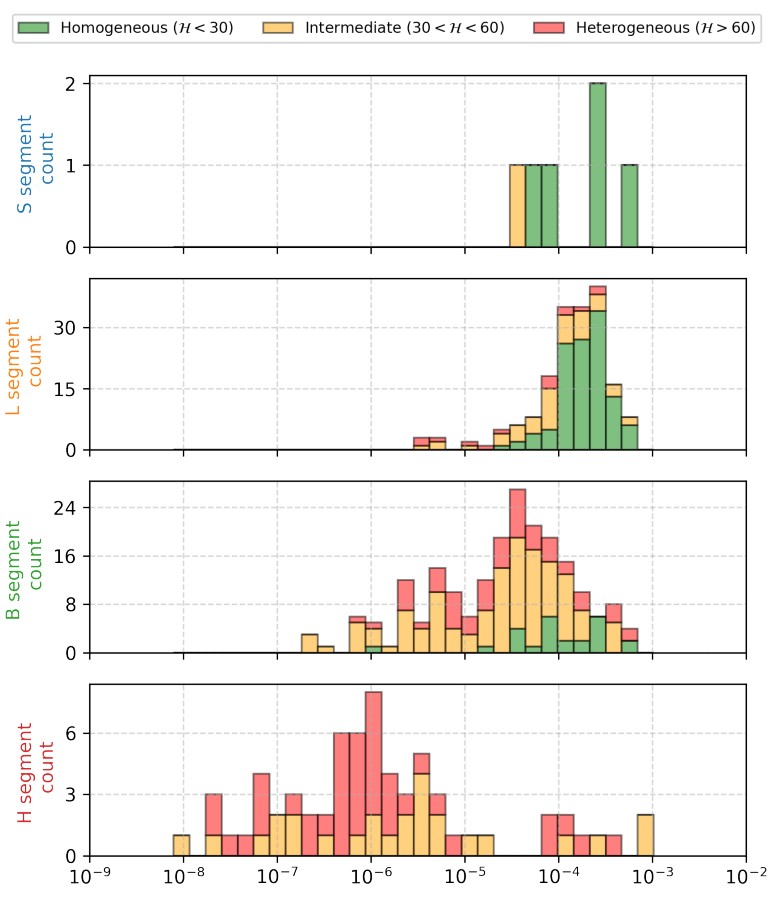

**Figure 17.** Histograms of TKE dissipation rates for the turbulence dataset.

users wanting to study the variability along the entire leg from turbulence to mesoscale, develop specific analyses, revise segmentation, or identify specific meteorological patterns or events in the time series, such as thermals or cool pools.

The nomenclature for the dataset file names is given in Table 3.

## 5    Conclusions

The article presents the MAESTRO turbulence dataset, based on the in-situ wind, temperature, and moisture measurements
from the instrumented ATR-42 from SAFIRE during the MAESTRO field campaign. It presented the sampling strategy and processing details. The quality of the variables of the dataset has been assessed, including the computation of systematic and random errors. On the one hand, the moment dataset can be used to perform process studies or model validation, as it



**Table 3.** File nomenclature for the MAESTRO turbulence dataset. *YYYYMMDD* corresponds to the flight date (e.g., 20240815), *NN* to the campaign flight number (e.g., 09), ZZ to the SAFIRE flight number identifier, *X* to the leg identifier and *i* the segment identifier (e.g., L2_1, the first segment of L2), and *V* the file version.

**Product**

**Moments**

MAESTRO_ORCESTRA_ATR_TURBULENCE_MOMENTS_RF*NN*_as2400ZZ_*YYYYMMDD*_v*V*.nc

**Fluctuations**

MAESTRO_ORCESTRA_ATR_TURBULENCE_FLUCTUATIONS_RF*NN*_as2400ZZ_L*X*_*i*_*YYYYMMDD*_v*V*.nc

**Leg scale calibrated time series**

MAESTRO_ORCESTRA_ATR_TURBULENCE_CALIBRATED_RF*NN*_as2400ZZ_L*X*_*YYYYMMDD*_v*V*.nc

summarizes the turbulence and mean data effectively in few key metrics that can be easily compared across segment types or flights. On the other hand, leg-scale time series dataset can be used to identify larger scales spatial variations that were

cut off by the filtering using the calibrated time series, and turbulent fluctuations enable finer analysis to reprocess the data, or perform specific analyses with various methods such as wavelet analysis. Providing both approaches ensures the best and broadest utilization of the turbulence data collected during the MAESTRO field campaign.

With the data quality analysis, a few comments and recommendations for users of this dataset can be formulated. For turbulence statistics, segments with a heterogeneity score below 30 can be considered homogeneous and should be prioritized.

Additionally, the treatment of FAST-WAVE data will be improved to better resolve frequencies higher than 25 Hz. A dedicated 100 Hz dataset will be released on the AERIS website following this publication. While being a valuable resource for the broad atmospheric research community, the present dataset was primarily collected to address the MAESTRO research questions. The data presented in this article will be used to examine the relationships between turbulence characteristics within the marine boundary layer and the organization of convection in the tropics. As one of several MAESTRO and ORCESTRA datasets, it will

be integrated with other observations to conduct analyses targeted at these research questions. Other datasets include sounding near Sal airport, airborne remote sensing instruments, satellite observations, and in-situ microphysics. Among the satellite products that were gathered during the campaign, SAR images are particularly promising. They document the centimeter-scale waves at the surface of the ocean (Atlas, 1994), thus they are relevant to the study of turbulence in the marine boundary layer and sea-air-cloud interactions (Brilouet et al., 2023; Kudryavtsev et al., 2005). Seven flights of the campaign were planned with

respect to these acquisitions so that the plane was concomitantly performing a full leg at 500 ft above the surface of the ocean. Exploring these data could lead to better interpretations of SAR images and plane turbulence data, and ultimately improve the current understanding of air-sea interactions within the tropical convective environment. This represents one of the direct research avenues for this dataset and a potential building block for further analyses, hopefully advancing the more fundamental questions that motivated MAESTRO and ORCESTRA.



## 6 Code and data availability

Code is availble on request to the corresponding authors. The turbulence dataset is available at https://doi.org/10.25326/812 (last access:24/09/2025, Jaffeux and Lothon (2025). The raw dataset used for this study is available on the AERIS website https://doi.org/10.25326/738 (last access:24/09/2025, Bony (2024)).

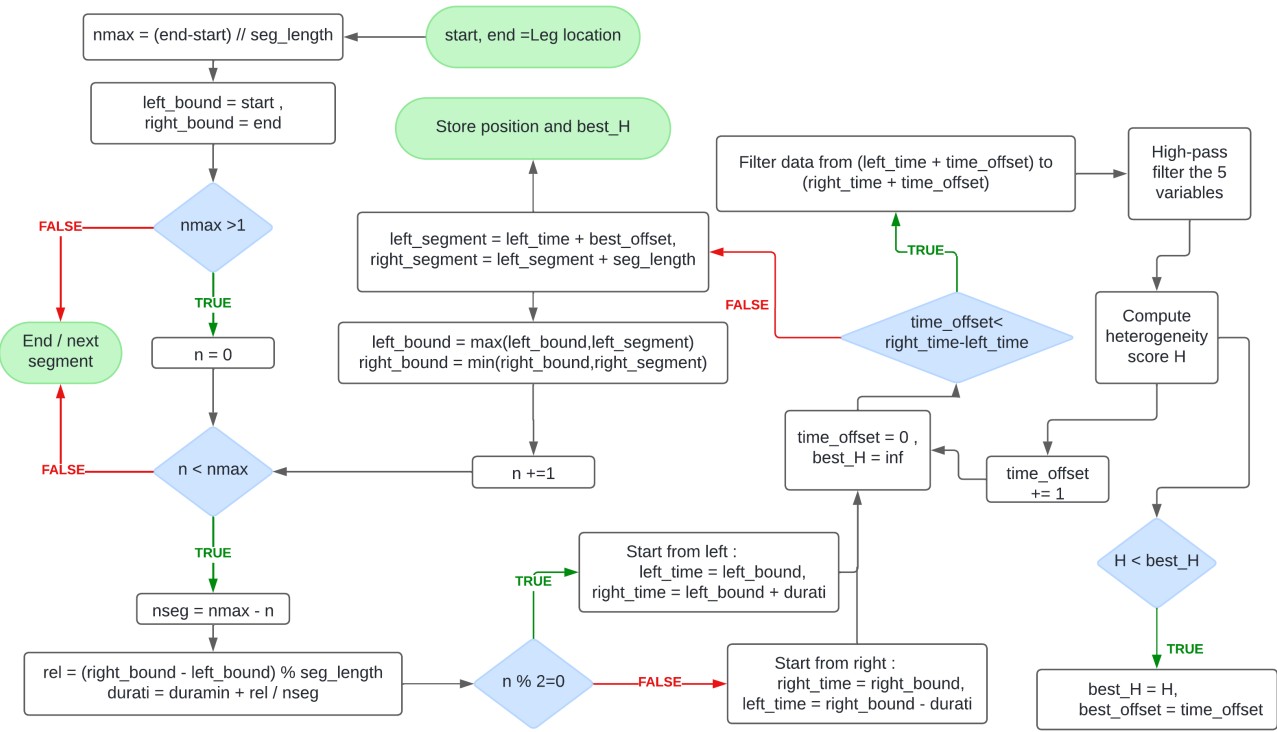

**Figure A1.** Segmentation algorithm flowchart

*Author contributions.* LJa wrote the article; LJa wrote the analysis code; ML and LJa analysed the data; ML, SB, FC, DB revised the manuscript. SAFIRE processing team (GC, CSL, HB, HO, DC, TJ) produced and curated the initial ATR 25Hz dataset which was used to produce the final turbulence dataset. JB and LJo adapted the FAST-WAVE hygrometer to be embarked on the ATR and curated its data. SB is the lead scientist and coordinator of the MAESTRO campaign.

*Competing interests.* The authors declare no competing interests.



*Acknowledgements.* ATR42 Airborne data were obtained using the aircraft managed by Safire, the French facility for airborne research,
an infrastructure of the French National Center for Scientific Research (CNRS), Météo-France, and the French National Center for Space
Studies (CNES). The authors gratefully acknowledge all the SAFIRE staff, technicians, engineers, pilots and directors for their considerable
efforts and involvement in the realization of the MAESTRO airborne operations. This research received funding from ERC ( MAESTRO
grant no. 101098063) and ESA (contract no. 281042).





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
