# Peer review of "The MAESTRO turbulence dataset derived from the SAFIRE ATR42 aircraft"

_Earth System Science Data, 2025_

## Referee Comment (RC1)

Review on

„The MAESTRO turbulence dataset derived from the SAFIRE ATR42 aircraft“

by Louis Jaffeux, Marie Lothon, Fleur Couvreux, Dominique Bouniol, Grégoire Cayez, Lilian Joly, Jérémie Burgalat, Cyrille De Saint Leger, Hubert Bellec, Olivier Henry, Dyaa Chbib, Tetanya Jiang, and Sandrine Bony

**General**

The manuscript describes a data set of atmospheric observations measured above Cabo Verde using the ATR42 research aircraft as part of the MAESTRO campaign. The basic meteorological parameters are stored with a relatively high temporal resolution of 25 Hz. In total, data sets from 24 measurement flights between August 10 and September 10, 2024, are available. Most of the data was measured on horizontal flight paths, with flight altitudes ranging from low altitudes of up to 60 m above sea level to 6000 m, with the sections at higher altitudes being used primarily for downward-facing remote sensing methods.

The focus is on the high temporal resolution with associated spatial resolution of a few meters, which also allows turbulence investigations. While the wind vector and air temperature were measured using the aircraft's standard instrumentation, the campaign provided an opportunity to use new, high-resolution humidity sensors.

After an introduction that essentially presents the overarching ORCHESTRA initiative and the various aircraft campaigns associated with it, the actual motivation behind the MAESTRO campaign is briefly presented. Although the spatial distribution of clouds and their microphysical properties were also mentioned as motivation for the campaign, this dataset focuses more on the sub-cloud layer and the connection to the air masses below.
The data set presented here does not contain any further information on cloud flights, which at first glance seems a little confusing given the motivation behind the campaign.

The second chapter presents the measurement campaign itself, the instrumentation, and the measurement strategy. However, I see major problems, particularly in the presentation of the instrumentation: in my opinion, the section lacks structure. Although reference is made to further literature, various details of the systems presented are selected and presented somewhat arbitrarily. Furthermore, it is very difficult to identify the sensors precisely, as they are often only mentioned using abbreviations without manufacturer details, etc. In my opinion, this chapter does not meet the standards for the introduction of instrumentation. I would have expected at least one sensor table with manufacturers, response time, and accuracy at this point.

Particularly with regard to the typical acquisition frequencies of the individual sensors, there is often no clear distinction between the actual sampling frequencies and the native temporal resolution, i.e., the response time. Unfortunately, I cannot get an impression of the quality and accuracy of the sensors from this section. For example, two capacitive humidity sensors are mentioned that are supposedly calibrated, but there is no indication of how and against which standard this was done.

The description of the flight strategy and the associated classification into "types" is somewhat unusual, but it is certainly possible to do it this way.

I am somewhat critical of the further subdivision of the legs into "homogeneous sections" with regard to the fluctuations of five selected parameters, and I am not sure what benefit this classification has for users of the data. I have the impression that this classification is only of interest to users who want to keep their own data analysis effort to a minimum. That is certainly legitimate, but it also limits the possibilities of data analysis somewhat. In my opinion, the explanation of the heterogeneity factor requires a few more clarifications (see detailed comments).

I see a general problem in the order of sections 2 and 3: Section 2 describes the division of observations according to flight patterns and homogeneous sections, but this is followed in section 3 by the more fundamental processing and calibration of the data. The order seems unfortunate to me, as it does not correspond to the actual order of post-processing. I suggest reconsidering this order.

My last major point relates to the "calibration" of the sensors for rapid measurements, which is explained in great detail and with many illustrations. It is certainly standard practice to combine fast but rather inaccurate and fragile sensors with comparatively slow but more robust and easier to calibrate sensors. However, I am not yet entirely convinced by the method proposed here.
First, there is no mention of whether the slower sensors are corrected for their inertia before they are correlated with the faster sensors. Secondly, as already noted, there is no information about the accuracy of the sensors used as a reference. And finally, I do not understand why it is of interest to perform calibration for the different types of legs – is it assumed that the sensors behave differently at 60 m ASL than at 300 m ASL?

What I find completely missing in this data overview paper is a meteorological or synoptic classification of the 24 flights. Under what conditions were the flights carried out? This information would be very helpful for further use of the data, especially with regard to the cloud situation.

In summary, I believe that this manuscript definitely needs major revisions, although the data set itself is of great interest to external users.

**More detailed major and minor comments**

**Abstract:**
I suggest to mention the true airspeed in addition to the frequency of the stored data.
line 9ff: please explain what is meant by "turbulent moments"; you probably mean the statistical moments of the probability density function of the individual parameters. Please use precise terminology throughout the manuscript.

**1 Introduction:**
Line 20: Why only absorbing solar radiation? What's about terrestrial irradiance and also reflection of solar irradiance?

Line 28/29: The statement is somewhat vague; above all, I would rather say that the warm ocean provides latent heat for convection, which is then also associated with turbulence.

Line 39 to 52: The transition from EUREC4A to the ORCHESTRA campaigns was quite complicated for me to read. In particular, the fact that ORCHESTRA is a combination of several measurement campaigns was mentioned rather late in the text, which I found somewhat confusing when reading it for the first time. Perhaps the section could be revised slightly to make it easier to understand.

Line 44/45: "This campaign follows …" What exactly do you mean here with "following"?

Line 57/58: "in-situ turbulent scales …" the scales are not turbulent, better:   " in-situ observations of typical turbulence scales and fluxes in the sub-cloud layer.." ?

**2 The MAESTRO field campaign: acquisition strategy**

**2.1 Description of the campaign**

Line 1-3: However, none of the three objectives for the MAESTRO experiment primarily requires high-resolution turbulence data, correct?

Line 85: what exactly do you mean with "stabilized legs" ? I have an idea what you probably mean, but it should be defined briefly in one sentence.

Figure 1: The labels are quite small and bright - but might be okay. Are really all 24 flight patterns included in Fig 1?

**2.2 Instrumentation for turbulence measurements**

Section 2.2 on instrumentation should be improved in general: on the one hand, it states that the entire data set was published with a temporal resolution of 25 Hz; in Sec 2.2, the individual sampling frequency is specified again for each sensor, which is not really relevant; much more interesting is the actual temporal resolution/response time of the individual sensors.
For example, if I provide the dew point mirror data at 25 Hz even though I know that it cannot resolve this frequency, this is important information for the user of the data. If, on the other hand, I learn that the LiCor has a response time of 50 ms —i.e., 20 Hz—but the signal is sampled at 50 Hz and then made available at 25 Hz, I wonder what I am supposed to do with the individual pieces of information.
Finally, two capacitive humidity sensors are mentioned; one is sampled at 1 Hz, the other at 40 Hz, but what is the actual response time of the sensors? On the one hand, I am missing important information here, and on the other hand, rather unimportant information is provided—this should be sorted out a little and consideration given to what is really of interest to the user.

Line 102: be consistent with the nomenclature of units in line 96 you write "$m.s^{-1}$" , here you write "°C/sec" which should be "$K.s^{-1}$", furthermore, it should be the "response" and not the "response time" which has unit of "s".

**2.3 Flight sampling strategy**

Why not describing all applied leg types together in the same way? In line 115 to 120 you describe three (major) types with bullets and the "S-type" later on in the text. And then in line 149/150 you define even more types. I suggest summarizing them and mentioning that some types were flown more frequently and others were more of an exception.

Line 154: "turbulent dataset" makes no sense, you probably mean a set of "turbulence data", same in the next line "5 turbulent fluctuations" makes no sense.

Equation 1: In my opinion, the explanation of the heterogeneity factor requires a few more clarifications, for example, what exactly the function $F$ means. I also don't quite understand why the inner integrals in the fraction go over $x$ and the outer integral then over time $t$? Did you come up with this parameter yourself or are there references for it?

Line 206/207 "…projecting the geographical wind components onto the mean wind at the segment scale …". I understand what you mean, but the sentence doesn't make sense. Please rephrase it.

Figure 5: panel c, label of y-axis: please delete the bracket; and about the title of panel c: "power density spectrum of water vapor mixing ratio" - not of an instrument

Equation 4: I think this equation requires some additional background information. For example, it would be important to know how large the error is if the spectrum (e.g., in Fig. 5c) does not have the theoretical slope of -5/3 that you assume here (but do not mention anywhere).

Line 230: sigma is a standard deviation or maybe you missed the power 2?

Line 231/232: I cannot follow this line of reasoning: where in Eq 4 can I see that the energy dissipation rate depends on the velocity gradients? And where does wind shear appear in the equation?

Line 233/234: The distributions of energy dissipation rates shown in Fig. 17 show very low values, especially in the H segment, and I have serious doubts that a measurement system on an aircraft can accurately resolve dissipation rates down to $10^{-8}$ $m^2s^{-3}$. Can you please estimate the measurement accuracy or maximum resolution?

**3.2 Pre-Processing**

The beginning of Section 3.2 is very confusing, as it appears to start with the introduction of a fast moisture sensor, but then leads indirectly to the issue of wind determination. This should be restructured somewhat, as the content does not really match the heading of Section 3.2.

**3.2.1 Wind Corrections**

Line 252 Am I correct in understanding that the wind vector in the earth-fixed system is derived from the combination of the 5-hole arrangement on the radome and an additional Pitot-static tube? Why? Does the central hole of the 5-hole arrangement take on the function of a Pitot tube? This should be clarified a little further.

Line 257: I don't quite see it that way: I also need the same, if not greater, accuracy in the measurements to determine the average wind vector. When calculating the fluctuations, average values are subtracted, and the difference in determining the horizontal wind during flight maneuvers, for example, will be significantly smaller.

Line 263: I think the initially guess is displayed in Fig 6a and not in 6b - correct?

Line 268: Please check the panels which might disagree with the figure caption

Line 274: Why do you assume it's a "sensor defect"? There could be many different causes: installation errors with the IMU, calibration errors, and so on. With "sensor defect," I would really suspect a broken sensor.

**3.2.2 Calibrations of temperature and humidity instruments**

Line 295ff: When describing the four steps, I don't quite understand the reference to the individual panels in Fig. 7. In the first step, for example, you mention low-pass filtering of all four moisture measurement time series and refer to panel 7a, which shows the unfiltered data—is that correct? The low-pass filtered data is shown in 7b. Does that match the description of the second step? And finally, in 7d, the "fast-wave" data is shown as corrected for offset and slope relative to the other three humidity sensors. However, the four time series in 7d still differ by at least one constant offset—what is the most likely solution?

In terms of the absolute accuracy of the humidity sensors, the dew point mirror should have the best quality (although I am not very familiar with the other sensors), but it has the problem of high temporal inertia. Why is this property not consistently exploited and at least the offset determined via the mean value? I cannot quite follow this description of the calibration and therefore have serious doubts as to whether this is the right way to obtain high-quality humidity data.

Line 315: The sentence doesn't make sense to me; you write something about fast calibration, but you mean the calibration of the data from the fast sensor—is that right?

Line 318 - 320: In summary, I still have some doubts about the calibration of the fast sensors. Either I don't understand the procedure correctly, or the method is flawed. Using the correlation of two sensor signals to say something about the suitability of one sensor as a reference is somewhat risky, as the properties of a reference sensor are completely ignored. But as I said, maybe I have misunderstood something here and you can quickly convince me that this is the right way to do it.

Line 323-326: I cannot understand the content of these sentences without further explanation. What do you mean by "resampled," for example? Would it help to have another illustration in which you explain the method using an example?

**3.3.3 Temperature**

In principle, the same comments regarding the calibration of humidity sensors also apply here for temperature. However, there is another important point to consider: especially at the cloud base, droplet impaction could occur, or are you flying below the cloud base so that this can be ruled out?

Line 340: What evidence do you have to support the advantages of the 5-micron sensor? These are very vague statements.

Line 350/351: Since a properly performed calibration should increase the performance and reliability of a sensor, I cannot understand this statement. And why does the "de-iced" sensor not require calibration?

**3.3.4 3D Wind**

In the chapter on determining the offsets for the slide and angle of attack, you wrote that once these have been determined, you assume constant values for the entire measurement campaign. Furthermore, these are probably installation errors or influences on the flow around the radome or similar. Why is the matter not settled with that? I don't understand the physical background as to why statistics are now being presented for uncorrected measurements in comparison to the corrected measurements—or have I misunderstood something? What exactly can I learn from Fig. 12?

Fig 11, caption: should be „temperature" instead of „humidiy" - right?

Line 365/366: Which theoretical predictions of Kolmogorov's classical theory did you examine besides the spectral slope of -5/3?

**3.3.7 Systematic error**

Line 400/401: I don't understand this sentence; why is information lost during high-pass filtering? Of course, the choice of method for defining the fluctuations is somewhat arbitrary, but no information is lost in this context. It is also unclear to me why determining the fluctuations by subtracting the linear trends is used as a reference, so to speak, and why equation 7 can be used to determine or estimate a systematic error.

**4 Data Summary**

The summary in Section 4, and especially Fig. 16, is of course highly simplified when averaging all 24 flights and showing the subdivision "only" for the leg types. This can be done, but in my opinion, the benefit is rather limited, as can be seen from the conclusions drawn from Fig. 16. It is not a new finding to emphasize that mechanical turbulence is strongest near the ground. Somewhat more surprising is the higher variance of the

thermodynamic parameters at the cloud base; but why should the strongest gradients of T and q be found there? A cloud base is not necessarily the place where one would classically expect to find an inversion (cf. line 421).

In my experience, the arguments described in section lines 419–423 are more characteristic of cloud tops than cloud bases. For example, a temperature inversion at the cloud base would prevent convection and thus make cloud development rather impossible. Of course, there are also decoupled cloud layers, but if that is what is meant here, it should also be substantiated by observations. The term "entrainment" is also more associated with the cloud top than with the cloud base.

The energy dissipation rates shown in Fig. 17 are really very low, and I have serious doubts that values below $10^{-6}$ $m^2 s^{-3}$ can be statistically significantly resolved with the instrumentation on the ATR42—I have previously seen such low values only with very sensitive turbulence measurement systems such as hot-wire anemometry.
If my assessment is wrong, that's not a problem, but then a thorough analysis of the data is required. In any case, the spectral noise floor of the wind measurements should be determined in order to be able to estimate a meaningful resolution for the dissipation rate (see, for example, Muschinski et al. in Boundary-Layer Meteorology 98: 219–250, 2001).